# Systemic Treatment with Nicotinamide Riboside Is Protective in Two Mouse Models of Retinal Ganglion Cell Damage

**DOI:** 10.3390/pharmaceutics13060893

**Published:** 2021-06-16

**Authors:** Xian Zhang, Nan Zhang, Micah A. Chrenek, Preston E. Girardot, Jiaxing Wang, Jana T. Sellers, Eldon E. Geisert, Charles Brenner, John M. Nickerson, Jeffrey H. Boatright, Ying Li

**Affiliations:** 1Atlanta VA Center for Visual and Neurocognitive Rehabilitation, Decatur, GA 30033, USA; xianzhang_csu@163.com (X.Z.); nan.zhang@emory.edu (N.Z.); preston.girardot@emory.edu (P.E.G.); jana.t.sellers@emory.edu (J.T.S.); 2Department of Ophthalmology, Second Xiangya Hospital of Central South University, Changsha 410011, China; 3Department of Ophthalmology, School of Medicine, Emory University, Atlanta, GA 30322, USA; micah.chrenek@emory.edu (M.A.C.); jiaxing.wang@emory.edu (J.W.); eldon.e.geisert@emory.edu (E.E.G.); litjn@emory.edu (J.M.N.); 4Department of Diabetes & Cancer Metabolism, City of Hope National Medical Center, Duarte, CA 91010, USA; cbrenner@coh.org

**Keywords:** nicotinamide riboside, retinal ganglion cell, neuroprotection, microbead, optic nerve crush, ERG

## Abstract

Glaucoma etiology often includes retinal ganglion cell (RGC) death associated with elevated intraocular pressure (IOP). However, even when IOP is managed well, disease can progress. It is thus important to develop therapeutic approaches that directly protect RGCs in an IOP-independent manner. Compromised nicotinamide adenine dinucleotide (NAD^+^) metabolism occurs in neurodegenerative diseases, including models of glaucoma. Here we report testing the protective effects of prophylactically systemically administered nicotinamide riboside (NR), a NAD^+^ precursor, in a mouse model of acute RGC damage (optic nerve crush (ONC)), and in a chronic model of RGC degeneration (ocular hypertension induced by intracameral injection of microbeads). For both models, treatment enhanced RGC survival, assessed by counting cells in retinal flatmounts immunostained for Brn3a+. In the ONC model, treatment preserved RGC function, as assessed by pattern electroretinogram, and suppressed retinal inflammation, as assessed by immunofluorescence staining of retinal fixed sections for glial fibrillary acidic protein (GFAP). This is the first study to demonstrate that systemic treatment with NR is protective in acute and chronic models of RGC damage. The protection is significant and, considering that NR is highly bioavailable in and well-tolerated by humans, may support the proposition of prospective human subject studies.

## 1. Introduction

Glaucoma is the leading cause of irreversible blindness globally [1,2]. There are multiple risk factors associated with glaucoma [3,4,5,6,7,8], with elevated intraocular pressure (IOP) and aging being two of the more prominent ones [9,10,11]. Elevated IOP damages the axons of retinal ganglion cells (RGCs) that make up the optic nerve, eventually causing RGC dysfunction and loss and resulting in blindness [2,12]. Glaucoma is commonly treated with topically-applied IOP-lowering drugs that are well tolerated and effective in managing IOP [11]. However, even when IOP is managed well, the disease can progress, resulting in significant vision loss. It is reported that more than 10% of patients become completely blind in at least one eye [3,12,13,14]. It is thus important to develop therapeutic approaches that directly protect RGCs in an IOP-independent manner.

Compromised nicotinamide adenine dinucleotide (NAD^+^) metabolism occurs in neurodegenerative diseases and may be causative [15]. NAD^+^ is an essential cofactor in various biological processes, including metabolism, aging, cell death, DNA repair, and gene expression [13,14]. NAD^+^ levels in CNS tissue in vivo and in cultured neuronal cells decline with age and in neurodegeneration [16,17]. NAD^+^ levels also decrease in models of glaucoma, suggesting that NAD^+^ is critical to RGC health [18,19]. Treatment with nicotinamide (NAM), a precursor of NAD^+^, increases circulating NAD^+^ and is protective in models of RGC degeneration [18,20,21,22] and is being tested in a glaucoma clinical trial [23]. The maximal efficacy in models was achieved only at doses that, by human equivalent dose (HED), may be above a good safety profile, and was not protective in a significant portion of the treated cohorts [24,25,26]. Thus, though maintaining NAD^+^ levels by treatment with precursors is potentially therapeutic, NAD^+^ precursors in addition to NAM should be considered.

Nicotinamide riboside (NR), another NAD^+^ precursor [27], is orally available to humans, commonly exists in milk, and is active as an oral agent at a daily dose of 400 mg/kg by supplementation into food [28,29,30]. A mouse and human study comparing NR and NAM reported superior bioavailability for NR for both species [31]. Incubation with NR inhibits axon degeneration elicited by axon transection in cultured neurons [32]. Also, intravitreal injections of NR protected against RGC axon loss induced by TNF, possibly by upregulating kinases specific to NR biosynthesis and SIRT1-mediated autophagic flux [33]. In our previous work, prophylactic systemic treatment with NR provided significant protection in mouse models of retinal degeneration, including light-induced retinal degeneration [34] and several retinitis pigmentosa models [35]. However, the protective effects of NR treatment in glaucoma or RGC damage models have not been explored.

Several rodent models of RGC damage have been developed to study the cellular and molecular mechanisms that underlie RGC degeneration, including optic nerve crush (ONC) and intracameral injection of microbeads [36,37,38]. In the ONC mouse model, the crush injury leads to acute RGC death [37]. This damage model has served as a useful preclinical model to study neuronal survival as it induces significant RGC death with little variability [37,38]. In the model of intracameral microbead injection, trabecular meshwork is clogged by microbeads, leading to hindered aqueous outflow, increased IOP, and potentially RGC death [36]. This damage model is developing into a useful preclinical model of moderately chronic elevated IOP with associated RGC and vision loss [36]. Here we report the effects of prophylactic systemic treatment with NR on RGC function and survival in those two models. NR treatment increased retinal NAD^+^ concentration, significantly preserved RGC numbers and function, and prevented ONC-induced reactive gliosis.

## 2. Materials and Methods

### 2.1. Animals

C57BL/6J mice were bred in-house and were 2–3 months of age when used in ONC experiments. The DBA/2J mice in this study were bought from The Jackson Laboratory (Stock No: 000671; Bar Harbor, ME). DBA/2J mice develop pigmentary dispersion glaucoma around 6–9 months of age, though initial iris defects may appear as early as 3–4 months of age [39,40,41,42,43]. DBA/2J mice were 6 weeks of age when used in intracameral microbead injection experiments, well prior to disease effects on RGCs. Both strains were maintained on a 12 h:12 h light: dark cycle with food and water provided ad libitum. All procedures involving animals were approved by the Animal Care and Use Committee of Emory University (PROTO201800248; 11/19/2018) and were in accordance with the ARVO Statement for the Use of Animals in Ophthalmic and Vision Research. As such, experiments met the 3R requirements and were conducted with adequate power.

### 2.2. Drug Treatments

NR chloride (NIAGEN; Item# ASB-00014332-101, Lot# 40C910-18209-21) was kindly provided by ChromaDex, Inc. (Los Angeles, CA, USA). The drug vehicle was phosphate-buffered saline diluted from a 10× stock (PBS; VWR, Cat# 97063-660), which has a 1× solution composition of 137 mM NaCl, 2.7 mM KCl, and 9.5 mM phosphate buffer (diluted to 1× working solution with Molecular Biology Water, Corning, LOT# 09516016, Manassas, VA, USA), pH 7.35. In experiments testing the effects of NR treatment in the ONC mouse model, NR (1000 mg/kg) was intraperitoneally (i.p.) injected into adult C57BL/6J mice 3 times per week. After 2 weeks, unilateral ONC was conducted followed by NR treatment the same day. A control cohort was identically treated with drug vehicle (PBS). In the microbead injection model, two treatments were administrated to DBA/2J mice before microbead injection. NR (1000 mg/kg) was injected i.p. once the day before and once the day of the microbead injection. After that, mice were i.p. injected with NR or PBS 3 times per week for 8 weeks.

### 2.3. Optic Nerve Crush (ONC)

The ONC procedure was performed unilaterally on C57BL/6J mice aged 2–3 months as we have reported previously [38]. The animals were deeply anesthetized with ketamine (10 mg/mL; AmTech Group Inc, Lake Forest, IL, USA) and xylazine (100 mg/mL; AKORN, Lake Forest, IL, USA), followed by topical application of 0.5% proparacaine hydrochloride ophthalmic solution (Falcon Pharmaceuticals, Fort Worth, TX, USA) and providone-iodine (VWR, RC3955-16, Radnor, PA, USA). Prior to crush, the eye was rinsed with refresh tears (Refresh, Lot# 94910, Irvine, CA, USA). Triple Antibiotic Ointment (McKesson, MFR3 118-42213, Richmond, VA, USA) was applied to both eyes to keep them moist during the procedure. Under a stereo surgical microscope, a cut was made at the lateral canthus of the eyelid to expose the lateral side of eyeball. Then, a small incision was made in the conjunctiva beginning inferior to the eyeball and around the cornea temporally, using micro-forceps to hold the edge of the conjunctiva next to the eyeball and retract it. The orbital muscles were gently deflected and the eyeball rotated nasally to expose the posterior aspect of the eyeball and optic nerve. The optic nerve was grasped about 1 mm from the back of eyeball for 5 s with self-clamping forceps (Dumont Tweezers N7; Electron Microscopy Sciences; Cat# 72864-D; Hatfield, PA, USA), with only pressure from the self-clamping action of the forceps to press on the nerve. The Dumont cross-action forceps were chosen because their spring action applies a constant and consistent force to the optic nerve. After 5 s, the optic nerve was released and the forceps were removed, allowing the eyeball to rotate back into place. A small amount of Triple Antibiotic Ointment was applied to protect the eyes from drying and inflammation. The mice were placed on a heating pad set to 37 °C and monitored carefully until fully recovered from anesthesia. For the first three days after the procedure, the mice were closely monitored for possible infection, bleeding, and loss of muscular control. Mice were euthanized 3 days or 7 days after ONC and eyes harvested for subsequent morphological analysis described below. A group of age-matched C57BL/6J mice that had no surgeries served as naïve controls.

### 2.4. Microbead Injection

A total of 60 DBA/2J female mice, 6 weeks old, received an injection of ferric microbeads into the anterior chamber of the left eye using an approach modified from Samsel et al., 2010 [44]. Briefly, magnetic microspheres (Spherotech PM-40-10, Lake Forest, IL, USA) with a diameter of 4.14 μm were washed in sterile 1× Hanks’ Balanced Salt Solution (HBSS, cat#14175095; Thermo Fisher Scientific; Waltham, MA, USA) 3 times to remove preservatives from the beads and diluted to half their original concentration (from 2.5% *w*/*v* to 1.25% *w*/*v*). The animals were deeply anesthetized with ketamine and xylazine. Approximately 10 μL of fully mixed bead solution was slowly injected into the anterior chamber of the left eyes of mice with a 30 G needle, delivering approximately 0.3 mg of beads into an eye. The needle was held in place while microspheres were pulled into the trabecular meshwork aided using a rare-earth ring magnet (Magcraft NSN0586, Home Depot, Atlanta, GA, USA). The magnet was preset on the eyeball and used to distribute the microspheres around the iridocorneal angle to reduce the outflow of aqueous humor via the trabecular meshwork. At the end of the surgery, Triple Antibiotic Ointment was applied to the cornea and mice were allowed to recover on a heating pad until fully awake. Right eyes served as non-injected controls. All DBA/2J mice were euthanized at 8 weeks after ocular injections.

### 2.5. Pattern Electroretinograms (pERG)

RGC function was assessed using pattern electroretinograms (pERG) and was taken as the amplitudes of the P1 and N2 components. P1 amplitude was measured from the N1 nadir to the peak of P1. N2 amplitude was measured from the preceding P1 peak to the nadir of N2 (similar to what is termed “pERG amplitude” [45] or “P1N2” [46]). Mice were dark-adapted overnight before pERGs were performed. The following day, mice were anesthetized with i.p. injections of 100 mg/kg ketamine and 15 mg/kg xylazine (ketamine; KetaVed from Vedco, Inc., Saint Joseph, MO; xylazine from AKORN, Lake Forest, IL, USA). Once anesthetized, proparacaine (1%; Akorn Inc., Lake Forest, IL, USA) and tropicamide (1%; Akorn Inc. Lake Forest, IL, USA) eye drops were administered to reduce eye sensitivity and dilate the pupils. Mice were placed on a heating pad (37 °C) under dim red light provided by the overhead lamp of the Celeris-Diagnosys system (Diagnosys, LLC, Lowell, MA, USA). The pattern stimulator was placed in contact with the eye; the flash stimulator for the contralateral eye acted as the reference electrode. Transient pERG responses were recorded using black and white vertical stimuli delivered on the Celeris system using manufacturer’s guidelines. Briefly, pattern stimuli of 50 cd·s/m^2^ were presented and 600 averaged signals with cut-off filter frequencies of 1 to 300 Hz were recorded under scotopic conditions. Each mouse was placed in its home cage on top of a heating pad (37 °C) to recover from anesthesia.

### 2.6. IOP Measurement

IOP measurements were taken during microbead injection experiments using a rebound tonometer (Tonolab Colonial Medical Supply, Londonderry, NH, USA) on mice anesthetized with 5% isoflurane (Forane, Cat# 803250, McKesson Medical Surgical Inc., Irving, TX, USA). IOP readings obtained with the Tonolab instrument have been shown to be accurate and reproducible in various mouse strains, including DBA/2J [47]. Data were averaged values of 18 repeated measurements per animal and time point (automatic triplicate sampling by the tonometer, 6 measurements taken by the user). IOP was measured one day prior to microbead injection (baseline) and 1, 3, 5, and 7 days after injection and then every 3 days until 60 days post-injection. The “days post-injection” was plotted on the x-axis and the “IOP value” was plotted on the y-axis. We came up with the concept of “IOP Area” to indicate the area under the curve (AUC) of IOP change, which takes into account IOP changes over the duration of an experiment. IOP Area was measured with ImageJ [48].This involved using “Polygon selections” to select the area we would like to measure, then clicking “Measure” under “Analyze”. The software gives as output an “area reading” of the selected area. An IOP higher than 50 mmHg may cause ischemic optic neuropathy, which differs from chronic glaucomatous optic neuropathy [49]. The IOP of the mock-treated mice maintained a normal range. We excluded two mice whose IOP values were above 50 mmHg in two consecutive measurements.

### 2.7. Retina Flatmount Preparation and Imagining

For retina flatmount preparation, mice were deeply anesthetized with ketamine and xylazine as detailed above and perfused through the heart with 0.9% saline (Thermo Fisher Scientific; Lot# 130586; Waltham, MA, USA) followed by 4% paraformaldehyde (Electron Microscopy Sciences; Lot# 191203-27; Hatfield, PA, USA). The dissected eyeballs were post-fixed in 4% paraformaldehyde for an extra hour at room temperature. Retinas were removed from the globe and cut into quarters, creating four flaps or petals. Then, retinas were rinsed in PBS with 1% Triton X-100 (Sigma-Aldrich, Lot. #SLBW6852, St. Louis, MO, USA) and blocked with 5% normal donkey serum in a 96-well plate, and placed in primary antibody, Brn3a^+^ (1:1000; Santa Cruz; SC-31984; Dallas, TX, USA), overnight at 4 °C. The whole mount retinas were washed in PBS for 3 times and then placed in secondary antibody (AlexaFluor 488 AffiniPure Donkey Anti-Goat, Invitrogen, Lot# 1979698, Eugene, OR, USA) at 1:1000. After 3 washes in PBS, cover slips were placed over the retina whole mounts using Fluoromount-G (Southern Biotech, Cat. # 0100-01, Birmingham, AL, USA).

Retinal flatmounts were imaged with a Nikon Eclipse Ti-C1 confocal microscope (Nikon, Inc., Melville, NY, USA). For each retina, four fields (636.5 μm × 636.5 μm each), whose distance to the optic nerve was equal, were acquired at each eccentricity of the tissue (dorsal, nasal, ventral, and temporal) from middle regions of retina under 20× magnification [50]. Images were processed with a custom pipeline in CellProfiler [51] to count the number of Brn3a+-labeled cells in a masked manner [52]. The cell counts from each of the four flaps per retina was averaged for statistical analysis. For statistical analysis, the immunopositive RGCs per field were compared.

### 2.8. Ocular Section Preparation and Imaging

Mice were euthanized after pERG measurement on the third day after ONC. Histologic procedures followed techniques as we described before [34,53]. Eyes were dehydrated, embedded in paraffin, and sectioned through the sagittal plane on a microtome at 5 µm increments. Sections containing the optic nerve and the center of the cornea were selected for staining to ensure that consistent regions were examined between animals. The slides were deparaffinized across five Coplin jars with 100 mL of xylene for 2 min each, consecutively. Then the slides were rehydrated in a series of 100 mL ethanol solutions for 2 min each: 100%, 90%, 80%, 70%, 60%, and 50%. The slides were immersed in PBS (137 mM NaCl, 2.7 mM KCl, 9.5 mM phosphate buffer, pH 7.35) for 5 min each.

For detecting glial fibrillary acidic protein (GFAP) in ocular sections, sections were blocked for 30 min in 0.1 M Tris-buffered saline (TBS; Corning, Lot# 02616007, Manassas, VA, USA) containing 5% normal donkey serum (US Biological, S1003-36, Salem, MA, USA), then incubated in the primary antibody (rabbit anti-GFAP; 1:500; Z0334; Agilent Dako, Santa Clara, CA, USA) diluted in the blocking serum overnight at 4 °C. Sections were rinsed three times with 1× TBS (20 M Tris, 136 mM NaCl, pH 7.45) for 15 min each following primary antibody incubation, then incubated for 2 hours at room temperature with the secondary antibody solution (donkey anti-rabbit IgG; 1:1000; Alexa Fluor 568; A10042; Life Technologies, Eugene, OR, USA). Sectionsf were then washed three times with TBS for 15 min each, mounted with a 4′,6-diamidino-2-phenylindole (DAPI) mounting medium (Sigma-Aldrich, LOT# SLCG2144, St. Louis, MO, USA), and coverslipped.

The slides immunostained for GFAP were imaged using a Nikon Ti2 inverted microscope with A1R-HD25 confocal scanner (Nikon Instruments Inc., Melville, NY, USA). The slides were imaged using a 20× objective lens, resonance scanning with 2× zoom at 1024 × 1024 per field, processed using Nikon’s Denoise.ai algorithm, and tiled using NIS Elements. GFAP quantification was achieved by counting GFAP-positive fibers that fully penetrated into the inner nuclear layer (INL) as others have reported [54].

### 2.9. NAD^+^ Measurements

Levels of NAD^+^ in homogenates were measured using a commercially available kit by following manufacturer’s instructions (Abcam; ab. 65348; #Lot: GR3226737-3; San Francisco, CA, USA). In brief, retina samples were homogenized in extraction buffer. Extracted samples were separated into two aliquots. One was used to measure total NAD (NADt). For NADH-specific measurements, samples were heated to 60 °C for 30 min to decompose NAD^+^, leaving only NADH remaining. Extracted samples were placed in a 96-well plate and the NADH developer was added into each well. The plate was placed into a hybrid reader (Synergy H1 Hybrid Reader, BioTek, Winooski, VT, USA) and read every 30 min at OD 450 nm while the color was developing. Data from the 2-h timepoint are presented. NADt and NADH concentrations were quantified against an NADH standard curve. NAD^+^ was calculated with the equation NAD^+^ = NADt − NADH.

### 2.10. Statistical Analyses and Masking

The personnel conducting assessments in experiments that required judgment were masked to the specific treatment group from which sampling arose. This included semi-automated marking of pERG peaks and nadirs, semi-automated counting of Brn3a-positive cells in retina flatmounts, and counting GFAP-positive fibers in fixed retina sections. Statistical analyses were conducted using Prism 8.4.2 Software (GraphPad Software Inc. La Jolla, CA, USA). One-way ANOVA with Newman-Keuls’ post-hoc test or Student’s *t*-test [55] were performed for biochemical and morphometric data. Unless otherwise noted, *n* is the number of animals per experimental group. For all analyses, results were considered statistically significant if *p* < 0.05. All graphs display data as mean ± SEM.

## 3. Results

### 3.1. NR Treatment Elevated Retinal NAD^+^ Levels

NR is a NAD^+^ precursor that when given systemically increases NAD^+^ levels in many tissues, including CNS structures [56,57]. As an initial test of whether this may be the case in NR-induced RGC protection, we i.p. injected mice and with either NR or PBS daily for 5 days. Retinas were harvested one to two hours after the last treatment and assayed for NAD^+^ content. Compared to control cohorts injected with PBS, NR treatment increased retinal NAD^+^ in DBA2J mice (45.84 ± 1.61 vs. 53.53 ± 1.78 pmol/µL, *p* < 0.01; Figure 1A) and in C57BL/6J mice (38.69 ± 5.80 vs. 78.86 ± 6.11 pmol/µL, *p* < 0.01; Figure 1B).

### 3.2. NR Treatment Preserved RGC Function Following ONC

RGC function was assessed by measuring pERG responses and was taken as mean amplitude of the P1 component and mean amplitude of the N2 component (Figure 2A). The P1 component is the initial positive deflection originating from RGCs as well as from outer retinal photoreceptor cells [46]. The mean P1 amplitude was measured as the difference between the average N1 component nadir to the average P1 component peak (Figure 2D). The N2 component is the negative deflection following the P1 component that originates from the inner retina and reflects the RGC function [46]. The mean N2 amplitude was calculated from the average P1 component peak to the average N2 component nadir (Figure 2E). At 3 days post-ONC, both the P1 and N2 components were significantly diminished compared to contralateral eye or naïve eye responses (ONC+PBS vs. naïve, P1: 3.82 ± 0.81 vs. 9.27 ± 0.50 uV, *p* < 0.0001; N2: −6.56 ± 1.25 vs. −13.12 ± 0.59, *p* < 0.001; ONC+PBS vs. contralateral, P1: 3.82 ± 0.81 vs. 9.26 ± 0.38 uV, *p* < 0.0001; N2: −6.56 ± 1.25 vs. −14.03 ± 0.67, *p* < 0.0001, *n* = 12 in naïve group, *n* = 18 in contralateral group, *n* = 11 in ONC+PBS group) (Figure 2B–E). However, P1 and N2 responses from mice treated with NR were significantly preserved (ONC+PBS vs. ONC+NR, P1: 3.82 ± 0.81 vs. 6.89 ± 0.0.84 uV, *p* < 0.01; N2: −6.56 ± 1.25 vs. −11.47 ± 1.36, *p* < 0.01; *n* = 13 in ONC+NR group) (Figure 2B–E), suggesting that NR treatment preserved RGC function significantly during early ONC-induced axonal degeneration.

### 3.3. NR Treatment Prevented RGC Loss Following ONC

Numerous Brn3a-positive cells were observed in retinal flatmounts imaged by fluorescent confocal microscopy (Figure 3A). These were presumed to be RGCs. The highest density of these RGCs was seen in the control naïve retina. There was a decrease in RGCs 3 days following ONC. Retinas from mice treated with NR did not exhibit this ONC-induced RGC loss (Figure 3A). Semi-automated, masked counting of Brn3a-positive cells revealed that flatmounts from mice that had been i.p. injected with PBS and had undergone ONC had significantly fewer RGCs (1454.1 ± 83.01 RGCs/field) than those of naïve mice 3 days after ONC (1840.3 ± 95.29 RGCs/field; *p* < 0.01, one-way ANOVA Newman-Keuls test) (Figure 3B). Conversely, flatmounts from mice that had been i.p. injected with NR and had undergone ONC had a mean number of RGCs statistically indistinguishable from naïve mice and significantly greater than the mice that underwent ONC but had been injected with PBS (1746 ± 30.59 RGCs/field; *p* < 0.05, one-way ANOVA Newman-Keuls test) (Figure 3B). Seven days after surgery there was additional loss of RGCs in ONC-treated mice, and no effect of NR treatment (data not shown). Overall, the data suggest that treating with NR delayed RGC loss caused by ONC at the early stage.

### 3.4. NR Treatment Prevented Increased GFAP Expression after ONC

As an independent measure of the response of the retina to injury, we examined the expression of GFAP in sections through the retina. Mice were euthanized 3 days after ONC and retinal sections were stained for GFAP, a marker for reactive Müller cells and astrocytes [58]. In ocular sections from mice that had not undergone ONC, GFAP staining largely localized near the ganglion cell layer (GCL) and less so in the inner nuclear layer (INL; Figure 4A). In eyes of mice treated with PBS and that had undergone ONC, heavy GFAP staining was observed in processes or fibers spanning from the GCL to the INL and outer nuclear layer (ONL; Figure 4B). This additional signal was much less in the ONC mice treated with NR (Figure 4C). GFAP-positive fibers fully penetrating into the INL were counted across entire retina sections. This quantification confirmed that GFAP expression was statistically significantly increased following ONC in mice treated with PBS but that this increase was not observed after ONC in mice treated with NR (naïve vs. ONC+PBS vs. ONC+NR: 27.67 ± 4.60 vs. 185.70 ± 42.37 vs. 62.83 ± 13.55, *p* < 0.01 for comparisons of ONC+PBS with the other two groups) (Figure 4D).

### 3.5. Elevation of IOP by Intracameral Injection of Microbeads

Five cohorts of DBA/2J mice were used in this experiment (see legend, Figure 5). The naïve cohort had no treatments at all. The “Buffer” cohort received no i.p. injections but received unilateral intracameral injections of microbead buffer, HBSS. The “NR only” and “NR+Microbeads” cohorts received multiple i.p. injections of NR (1000 mg/kg) and the “PBS+Microbeads” cohort received multiple i.p. injections of PBS, with the latter two cohorts also receiving unilateral anterior ocular injections of microbeads per the schedule detailed in Methods.

IOP was measured at multiple days for each eye and averaged across each eye-specific treatment (Figure 5A). Compared to eyes not injected with microbeads, nearly all of the microbead-injected eyes showed IOP elevation, of which 83% elevated to above 20 mmHg. The average IOP of the microbead-injected eyes rose quickly on the second postoperative day, peaked at 8 days, and stayed elevated throughout the following 2–6 weeks, though NR treatment maintained elevation of IOP more so than PBS treatment (Figure 5A). The area under the curve (AUC) of IOP change, which takes into account IOP changes over the duration of an experiment (Figure 5B), was used as a reflection of ocular hypertension (OHT) [49]. The AUC significantly increased in both groups that received ocular microbead injections (111,325 ± 5060 IOP AUC for PBS+Microbeads group and 77,892 ± 1089 for NR+Microbeads group) compared to groups that did not receive any ocular injections (81,389 ± 695.4 for Naïve group and 77,892 ± 1089 for NR only group) or received ocular injection of buffer (73,836 ± 1083 for Buffer group; *p* < 0.05 or less for means of either microbead-injected group versus non-microbead-injected groups) (Figure 5C). Thus, this method proved to be a reliable approach to acutely elevate IOP in the mouse eye.

### 3.6. Systemic NR Treatment Prevented Loss of RGCs Caused by Intracameral Injection of Microbeads

Brn3a-positive cells observed in retinal flatmounts imaged by fluorescent confocal microscopy were considered RGCs (Figure 6A). Semi-automated, masked counting of these cells revealed that for naïve mice or mice systemically injected with PBS, RGC numbers were significantly lower in eyes that were intracamerally injected with microbeads (1566 ± 40.54 mean RGCs/field; red) compared to naïve eyes (1848 ± 57.3 mean RGCs/field; black) or eyes injected with microbead buffer (1912 ± 65.32 mean RGCs/field; gray). Conversely, for mice systemically injected with NR, intracameral injection of microbeads did not affect RGC numbers (1905 ± 39.36 mean RGCs/field; green), suggesting that systemic NR treatment prevented RGC loss (Figure 6B).

### 3.7. Correlation between Microbead-Induced IOP Changes and RGCs Survival Is Precluded by NR Treatment

As shown in Figure 5, the IOP AUC significantly increased in both groups that received ocular microbead injections compared to groups that did not receive any ocular injections or received ocular injection of buffer. Of note, systemic NR treatment increased IOP AUC compared to systemic PBS treatment (green versus red bar, Figure 5C). There was a significant negative correlation with RGC survival and IOP AUC for the eyes injected with microbeads in mice i.p. injected with PBS (“PBS+Microbeads” treatment group); as IOP AUC increased, the number of RGCs per flatmount field declined (r = −0.48, *p* = 0.003; Figure 7). This was not the case for eyes injected with microbeads in mice i.p. injected with NR (r = 0.11, p 0.524). A Chow test [59] for differences in regression coefficients indicates the two coefficients are statistically significantly different (F = 11.87, *p* = 0.000037). These data and analyses show that OHT induced by microbead injection correlates with loss of RGCs, but that this correlation is absent with NR treatment, suggesting that NR treatment might make RGCs less vulnerable to damage caused by OHT.

## 4. Discussion

NAD^+^ metabolism is dysregulated in several models of neurodegenerative disease, including axonal degeneration diseases [60] and in models of glaucoma [18]. Thus, maintaining NAD^+^ levels may be useful in preventing neurodegenerative diseases. Prophylactic systemic treatment with NR, a NAD^+^ precursor, is protective in several mouse models of neurodegenerative disease, including Alzheimer disease [56], Parkinson disease [57], and retinal degenerations [34,35]. We report here that systemic delivery of NR significantly increased NAD^+^ levels in retinas of C57BL/6J and DBA/2J mice (Figure 1), similar to findings we reported in BALB/c mice [34,35]. The difference between same-strain cohorts of levels of retinal NAD^+^ is considerably different in the two strains, with an increase of ~17% for DBA/2J and an increase of ~205% for C57BL/6J mice. As the C57BL/6J and DBA/2J mice were treated identically, this distinction may be due to strain differences.

To determine whether this increase of retinal NAD^+^ can protect against RGC damage and degeneration, we tested the effect of NR treatment in both acute and chronic mouse models of RGC damage. ONC, an acute glaucoma mouse model, reliably resulted in diminution of RGC function (Figure 2) and number (Figure 3). NR supplementation significantly delayed RGC dysfunction and RGC loss at an early stage (3 days) after ONC (Figure 2 and Figure 3). Intracameral microbead injection resulted in sustained IOP elevation and RGC loss spanning two months (Figure 5 and Figure 6) and the two outcomes were correlated (Figure 7), mimicking prevalent forms of glaucoma [49,61]. NR treatment prevented RGC loss across the two-month duration of the experiment and uncoupled the correlation of RGC loss to AUC of IOP change (Figure 5, Figure 6 and Figure 7), indicating that NR treatment protected RGCs in a relatively chronic model of glaucoma. It may be that systemic NR treatment led to elevation of NAD^+^ in RGCs, making them resilient to metabolic stress and oxidative damage caused by OHT and suppressed axonal transport [18,20] or autophagic flux via upregulation of SIRT1 [33].

Retinal GFAP immunosignal increased following ONC, suggesting activation of Müller glia cells [58]. This was not observed in mice treated with NR (Figure 4). These effects may be secondary to RGC cell-autologous mechanisms, though it is also possible that NR treatment increased intracellular NAD^+^ in Müller cells, which could suppress proinflammatory gene expression [62,63,64,65,66] known to be toxic to RGCs [67,68]. These and other mechanisms of NR-induced RGC protection are subjects of our future research.

NR treatment unexpectedly prolonged IOP elevation in eyes that were intracamerally injected with microbeads (Figure 5C), though these eyes showed preserved RGC numbers compared to microbead-injected eyes of PBS-treated mice (Figure 6). NR treatment had no effect on IOP in eyes of naïve mice or in eyes injected with buffer rather than microbeads (Figure 5C). Though conjecture, it is possible that NR treatment altered the responses of trabecular meshwork or juxtacannicular cells to microbead injection, possibly causing prolonging bead-induced suppression of aqueous humor outflow.

In summary, this is the first study to demonstrate that prophylactic systemic treatment with NR is protective in acute and chronic mouse models of RGC damage. The protection is significant, and may support the proposition of prospective human subject studies.

## Figures and Tables

**Figure 1 pharmaceutics-13-00893-f001:**
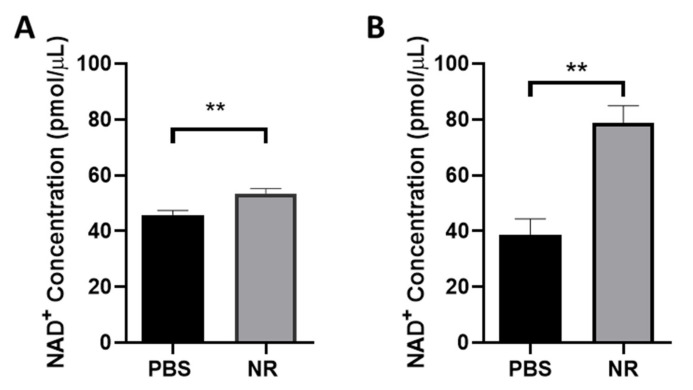
NR treatment increased retinal NAD^+^ levels in DBA/2J mice and C57BL/6J. Mice were i.p. injected with 1000 mg/kg NR or PBS for 5 consecutive days. One to two hours after the last injection, mice were euthanized and retinas harvested. NAD^+^ was measured by colorimetric assay. Retinal NAD^+^ levels from mice treated with NR were statistically significantly increased compared to vehicle (PBS) treated group. (**A**). NAD^+^ level comparison with and without NR treatment in DBA/2J mice. (**B**). NAD^+^ level comparison with and without NR treatment in C57BL/6J mice. ** *p* < 0.01 by Student’s t-test. N = 7–10 retinas/group. Error bars represent SEM.

**Figure 2 pharmaceutics-13-00893-f002:**
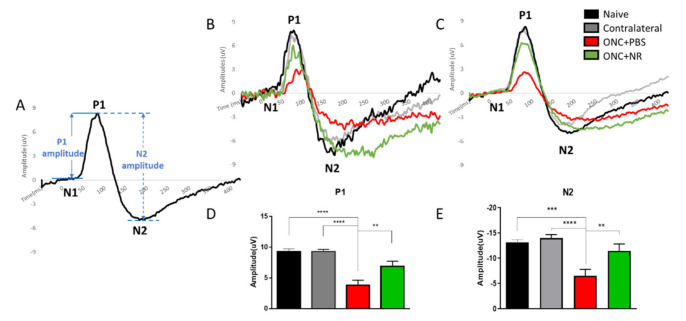
NR treatment preserved RGC function following ONC. (**A**). Representative example of pERG waveform and measurements of P1 and N2 amplitudes. (**B**). Representative pERG waveforms of one eye of a single mouse from each treatment group. (**C**). Composite pERG waveforms of all eyes from each treatment group. (**D**). Quantification of P1 mean amplitudes of the four treatment conditions. P1 mean amplitude of mice that underwent ONC and had been i.p. injected with PBS (red) was significantly diminished compared to the means of contralateral (gray) or naïve (black) cohorts. This diminution was partially but significantly prevented in mice that underwent ONC but had been i.p. injected with NR (green). (**E**). N2 mean amplitude of mice that underwent ONC and had been i.p. injected with PBS (red) was significantly diminished compared to the means of contralateral (gray) or naïve (black) cohorts. This diminution was prevented in mice that underwent ONC that had been i.p. injected with NR (green). ** *p* < 0.01, *** *p* < 0.001, **** *p* < 0.0001, one-way ANOVA with post-hoc Newman-Keuls test. N = 11–18 mice per group. Error bars represent SEM.

**Figure 3 pharmaceutics-13-00893-f003:**
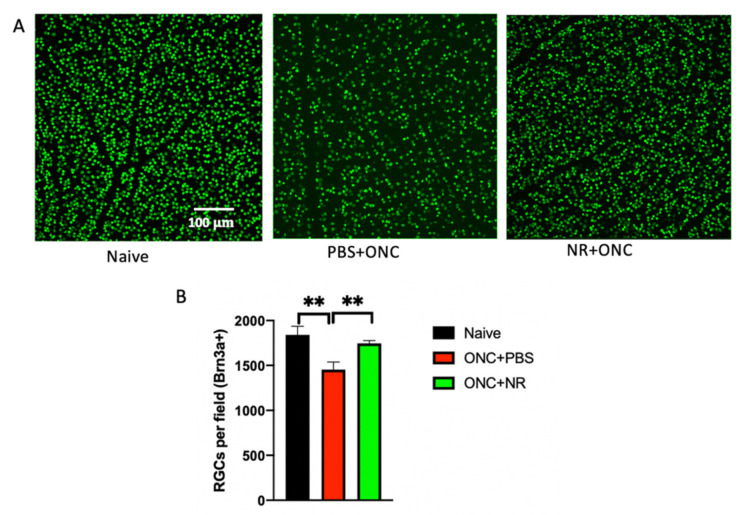
NR treatment prevents RGC loss following ONC. (**A**). Representative images of fields of retinal flatmounts stained for Brn3a+ from each cohort; green fluorescent signal is Brn3a+ immunosignal and is assumed to label RGCs. Mice in the “Naïve” group did not receive ONC or any treatment; the mice in “PBS+ONC” group were i.p. injected with PBS and underwent ONC; the mice in “NR+ONC” group were i.p. injected with NR and underwent ONC. (**B**). Quantification of cells stained with Brn3a+. Retinal flatmounts were prepared from mice of the three cohorts and RGCs were counted in 4 fields per flatmount using a semi-automated, masked protocol. The mean number of RGCs from mice that had been PBS-injected and had undergone ONC (red) were significantly lower than that of the naïve cohort (black). Conversely, the mean number of RGCs from mice that had been NR-injected and had undergone ONC (green) was statistically indistinguishable from that of the naïve cohort (black) and was significantly greater than the mean of mice that had been PBS-injected and underwent ONC (red), indicating that NR treatment prevented RGC loss at 3 days post-ONC. ** *p* < 0.01, one-way ANOVA Newman-Keuls test. N = 4–6 mice/group. Error bars represent SEM. Scale bar represents 100 µm.

**Figure 4 pharmaceutics-13-00893-f004:**
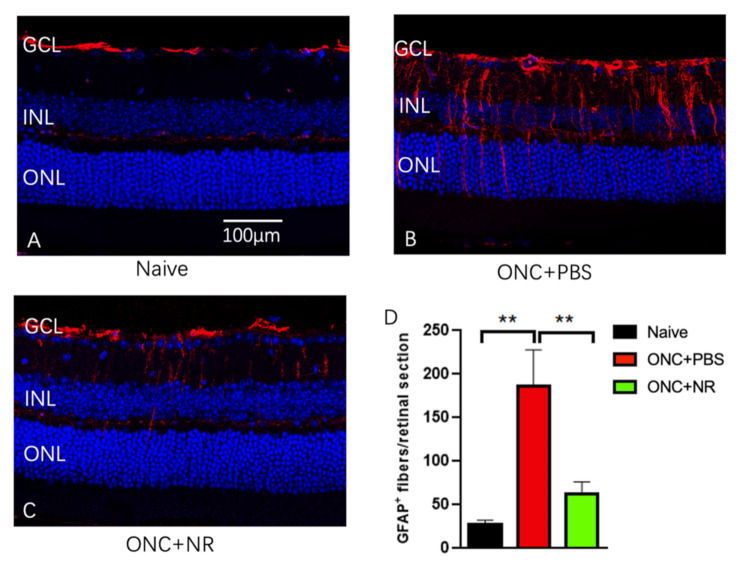
NR treatment prevents GFAP accumulating after ONC. Both PBS- and NR-treated mice underwent ONC and were euthanized at 3 days after ONC. (**A**–**C**). Representative morphologic images of each group; red is GFAP immunofluorescence signal and blue is DAPI staining. (**D**). GFAP-positive fibers fully penetrating into the INL were counted across the entire retina. NR-treated mice (green bar) exhibited significantly less GFAP signals compared with the PBS-treated group (red bar). GCL: ganglion cell layer, INL: inner nuclear layer, ONL: outer nuclear layer. ** *p* < 0.01 vs. other two groups, by one-way ANOVA with Newman-Keuls multiple comparisons post hoc test. N = 6 mice per group. Error bars represent SEM. Scale bar = 100 µm.

**Figure 5 pharmaceutics-13-00893-f005:**
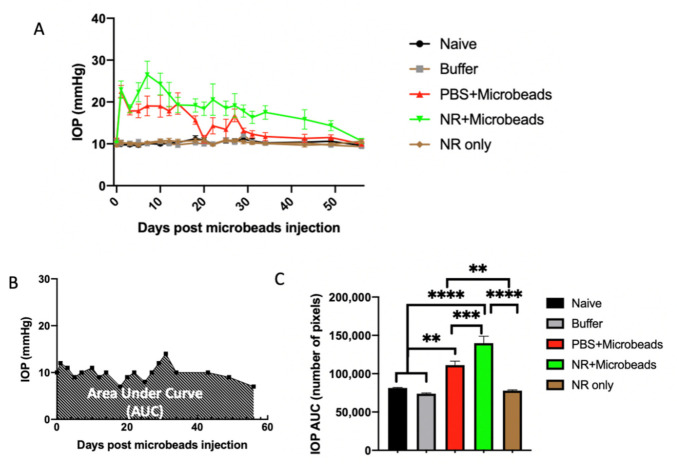
IOP changes over 2 months following microbead injection. (**A**). IOP was measured at the days given on the X-axis and averaged across all mice in each group. (**B**). Area Under the Curve (AUC) of IOP change was quantified by pixel counting using imageJ. This is an example of IOP AUC assessment on an individual eye from Naïve mouse. (**C**). Mean AUC of IOP change for each treatment group. The mice in the “Naïve” group (N = 10) did not receive microbead injection or any treatment; the mice in the “Buffer” group (N = 10) received buffer (HBSS) injection without any treatment; the mice in the “PBS+Microbeads” group (N = 19) received microbead injection and PBS i.p. injection; the mice in the “NR+Microbeads” group (N = 18) received microbead injection and NR treatment; the mice in the “NR only” group (N = 10) received NR treatment without anterior chamber injection. Microbead injection caused significant IOP AUC increases (compare red and green bars to black and gray bars). NR treatment in microbead-injected eyes (green bar) increased AUC IOP compared to PBS treatment (red bar). ** *p* < 0.01, *** *p* < 0.001, **** *p* < 0.0001 by one-way ANOVA with Newman-Keuls post-hoc test. Error bars represent SEM.

**Figure 6 pharmaceutics-13-00893-f006:**
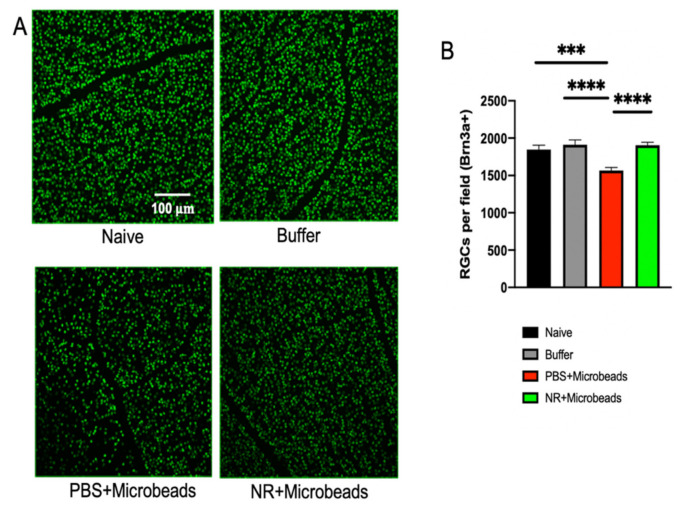
NR treatment prevents RGC loss following microbead injection. (**A**). Representative images of retina flatmounts immunostained for Brn3a+ from each cohort; green is Brn3a+ immunosignal and is presumed to stain RGCs. Treatment groups are as described in Figure 5 legend. (**B**). Mean number of RGCs per field for each treatment group. RGC numbers were significantly lower in mice systemically injected with PBS and intracamerally injected with microbeads (red) compared to naïve eyes (black) or eyes injected with microbead buffer (gray). Conversely, mice systemically injected with NR and intracamerally injected with microbeads (green) had a mean number of RGCs significantly greater than mice that had undergone ONC but had been injected with PBS (red). *** *p* < 0.001, **** *p* < 0.0001 by one-way ANOVA Newman-Keuls test. N = 10–19 mice/group. Error bars represent SEM.

**Figure 7 pharmaceutics-13-00893-f007:**
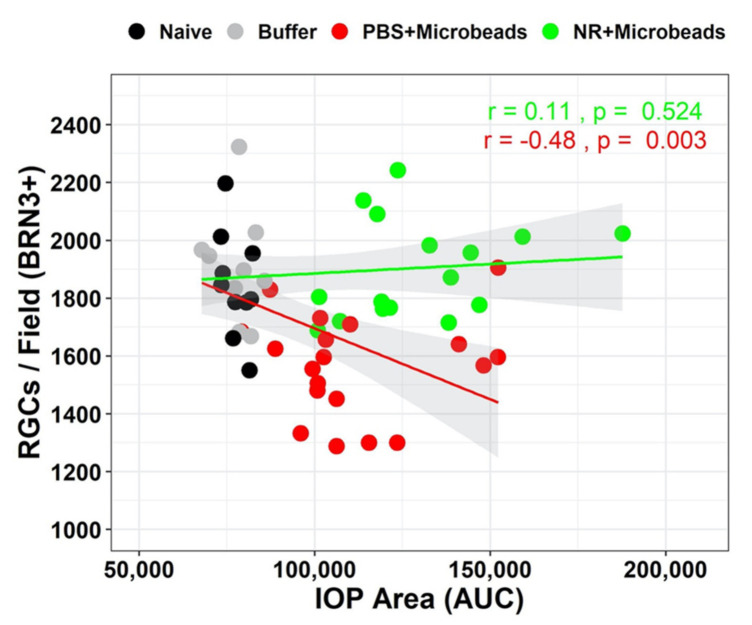
Correlation between Area Under the Curve of IOP changes and RGC survival. The mice in “Naïve” group did not receive microbead injection or any treatment; the mice from “Buffer” group received buffer (HBSS) injection without any treatment; the mice in “PBS+Microbeads” group received microbead injection and PBS i.p. injection; the mice in “NR+Microbeads” group received microbead injection and NR treatment. In PBS treatment group (red dots), BRN3a+ RGC number was negatively correlated with IOP AUC (r = −0.48, p = 0.003, Pearson correlation test); while in NR treatment group (green dots), as AUC increasing after microbeads injection, BRN3a+ RGC number did not show correlation (r = 0.11, *p* = 0.524, Pearson correlation test). A Chow test for differences in regression coefficients indicates the two coefficients are statistically significantly different (F = 11.87337, *p* = 0.000037). N = 10–19 per group.

## Data Availability

All data are contained within the submitted manuscript or supplementary material. The data presented in this study are available in this submission.

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
