# Peer review of "Systemic Treatment with Nicotinamide Riboside Is Protective in Two Mouse Models of Retinal Ganglion Cell Damage"

_pharmaceutics, 2021, doi:10.3390/pharmaceutics13060893_

Round 1
Reviewer 1 Report
In this work the authors obtained additional data indicating the relevance of NAD+ bioavailability to the development of glaucoma in two mice strains by using different methods (optic nerve crush and ocular hypertension) to produce damage to retinal ganglion cells. The animals received different schemes of 1g/kg doses of NAD+ systemically by intraperitoneal route, and then several aspects related to retinal function were assessed in non-treated and treated animals. The results confirmed previous studies by the authors and other groups indicating that NAD+ has a protective effect on retinal ganglion cells and suggested that systemic administration of VitaminB3 could have a therapeutic effect in glaucoma.
Major points requiring attention:
1) the authors should explain the reason why, in the ONC model, a dose of NAD+ was given to the mice one day prior to the procedure. Wouldn’t the damage to retinal ganglion cells be greater if the animals had not receive this ?
2) figure 1 shows that after the same 5 days NAD+ pretreatment, the levels of NAD+ in retinal homogenates are considerably different for the two mouse strains, with increase of only ~17% for DBA2J and ~205% for C57BL/6J. Is there an explanation for that? The authors should at least mention this finding in the discussion section.
Minor points:
-lines 302 and 305: correct typo…. through
-line 369-370:….ILM…. OPL…correct the abbreviatures to match those shown in the figure
-line 404: please rephrase, as it is not clear what the authors meant to say
-lines 404: what does OHT stand for? ocular hypertension ? it should be defined it at its first mention
-412: correct typo …. versus
A list of relevant articles not cited in the manuscript follows:
FN Clarivate Analytics Web of Science
VR 1.0
AF Cimaglia, Gloria
Votruba, Marcela
Morgan, James E.
Andre, Helder
Williams, Pete A.
TI Potential Therapeutic Benefit of NAD(+)Supplementation for Glaucoma and
Age-Related Macular Degeneration
SO NUTRIENTS
AB Glaucoma and age-related macular degeneration are leading causes of irreversible blindness worldwide with significant health and societal burdens. To date, no clinical cures are available and treatments target only the manageable symptoms and risk factors (but do not remediate the underlying pathology of the disease). Both diseases are neurodegenerative in their pathology of the retina and as such many of the events that trigger cell dysfunction, degeneration, and eventual loss are due to mitochondrial dysfunction, inflammation, and oxidative stress. Here, we critically review how a decreased bioavailability of nicotinamide adenine dinucleotide (NAD; a crucial metabolite in healthy and disease states) may underpin many of these aberrant mechanisms. We propose how exogenous sources of NAD may become a therapeutic standard for the treatment of these conditions.
PD SEP
PY 2020
VL 12
IS 9
AR 2871
DI 10.3390/nu12092871
ER
AF Chou, Tsung-Han
Romano, Giovanni Luca
Amato, Rosario
Porciatti, Vittorio
TI Nicotinamide-Rich Diet in DBA/2J Mice Preserves Retinal Ganglion Cell
Metabolic Function as Assessed by PERG Adaptation to Flicker
SO NUTRIENTS
AB Flickering light increases metabolic demand in the inner retina. Flicker may exacerbate defective mitochondrial function in glaucoma, which will be reflected in the pattern electroretinogram (PERG), a sensitive test of retinal ganglion cell (RGC) function. We tested whether flicker altered the PERG of DBA/2J (D2) glaucomatous mice and whether vitamin B3-rich diet contributed to the flicker effect. D2 mice fed with either standard chow (control,n= 10) or chow/water enriched with nicotinamide (NAM, 2000 mg/kg per day) (treated,n= 10) were monitored from 3 to 12 months. The PERG was recorded with superimposed flicker (F-PERG) at either 101 Hz (baseline) or 11 Hz (test), and baseline-test amplitude difference (adaptation) evaluated. At endpoint, flat-mounted retinas were immunostained (RBPMS and mito-tracker). F-PERG adaptation was 41% in 3-month-old D2 and decreased with age more in control D2 than in NAM-fed D2 (GEE,p< 0.01). At the endpoint, F-PERG adaptation was 0% in control D2 and 17.5% in NAM-fed D2, together with higher RGC density (2.4x), larger RGC soma size (2x), and greater intensity of mitochondrial staining (3.75x). F-PERG adaptation may provide a non-invasive tool to assess RGC autoregulation in response to increased metabolic demand and test the effect of dietary/pharmacological treatments on optic nerve disorders.
PD JUL
PY 2020
VL 12
IS 7
AR 1910
DI 10.3390/nu12071910
ER
PT J
AF Nzoughet, Judith Kouassi
de la Barca, Juan Manuel Chao
Guehlouz, Khadidja
Leruez, Stephanie
Coulbault, Laurent
Allouche, Stephane
Bocca, Cinzia
Muller, Jeanne
Amati-Bonneau, Patrizia
Gohier, Philippe
Bonneau, Dominique
Simard, Gilles
Milea, Dan
Lenaers, Guy
Procaccio, Vincent
Reynier, Pascal
TI Nicotinamide Deficiency in Primary Open-Angle Glaucoma
SO INVESTIGATIVE OPHTHALMOLOGY & VISUAL SCIENCE
AB PURPOSE. To investigate the plasma concentration of nicotinamide in primary open-angle glaucoma (POAG).
METHODS. Plasma of 34 POAG individuals was compared to that of 30 age-and sex-matched controls using a semiquantitative method based on liquid chromatography coupled to high-resolution mass spectrometry. Subsequently, an independent quantitative method, based on liquid chromatography coupled to mass spectrometry, was used to assess nicotinamide concentration in the plasma from the same initial cohort and from a replicative cohort of 20 POAG individuals and 15 controls.
RESULTS. Using the semiquantitative method, the plasma nicotinamide concentration was significantly lower in the initial cohort of POAG individuals compared to controls and further confirmed in the same cohort, using the targeted quantitative method, with mean concentrations of 0.14 mu M (median: 0.12 mu M; range, 0.06-0.28 mu M) in the POAG group (-30%; P = 0.022) and 0.19 mu M (median: 0.18 mu M; range, 0.08-0.47 mu M) in the control group. The quantitative dosage also disclosed a significantly lower plasma nicotinamide concentration (-33%; P = 0.011) in the replicative cohort with mean concentrations of 0.14 mu M (median: 0.14 mu M; range, 0.09-0.25 mu M) in the POAG group, and 0.19 mu M (median: 0.21 mu M; range, 0.09-0.26 mu M) in the control group.
CONCLUSIONS. Glaucoma is associated with lower plasmatic nicotinamide levels, compared to controls, suggesting that nicotinamide supplementation might become a future therapeutic strategy. Further studies are needed, in larger cohorts, to confirm these preliminary findings.
PD JUN
PY 2019
VL 60
IS 7
BP 2509
EP 2514
DI 10.1167/iovs.19-27099
ER
PT J
AF Williams, Pete A.
Harder, Jeffrey M.
Foxworth, Nicole E.
Cardozo, Brynn H.
Cochran, Kelly E.
John, Simon W. M.
TI Nicotinamide and WLDS Act Together to Prevent Neurodegeneration in
Glaucoma
SO FRONTIERS IN NEUROSCIENCE
AB Glaucoma is a complex neurodegenerative disease characterized by progressive visual dysfunction leading to vision loss. Retinal ganglion cells are the primary affected neuronal population, with a critical insult damaging their axons in the optic nerve head. This insult is typically secondary to harmfully high levels of intraocular pressure (IOP). We have previously determined that early mitochondria' abnormalities within retinal ganglion cells lead to neuronal dysfunction, with age-related declines in NAD (NAD(+) and NADH) rendering retinal ganglion cell mitochondria vulnerable to IOP-dependent stresses. The Wallerian degeneration slow allele, Wld(S), decreases the vulnerability of retinal ganglion cells in eyes with elevated IOP, but the exact mechanism(s) of protection from glaucoma are not determined. Here, we demonstrate that Wld(S) increases retinal NAD levels. Coupled with nicotinamide administration (an NAD precursor), it robustly protects from glaucomatous neurodegeneration in a mouse model of glaucoma (94% of eyes having no glaucoma, more than Wld(S) or nicotinamide alone). Importantly, nicotinamide and Wld(S) protect somal, synaptic, and axonal compartments, prevent loss of anterograde axoplasmic transport, and protect from visual dysfunction as assessed by pattern electroretinogram. Boosting NAD production generally benefits major compartments of retinal ganglion cells, and may be of value in other complex, age-related, axonopathies where multiple neuronal compartments are ultimately affected.
PD APR 25
PY 2017
VL 11
AR 232
DI 10.3389/fnins.2017.00232
ER
PT J
AF Masihzadeh, Omid
Ammar, David A.
Lei, Tim C.
Gibson, Emily A.
Kahook, Malik Y.
TI Real-time measurements of nicotinamide adenine dinucleotide in live
human trabecular meshwork cells: Effects of acute oxidative stress
SO EXPERIMENTAL EYE RESEARCH
AB The trabecular meshwork (TM) region of the eye is exposed to a constant low-level of oxidative insult. The cumulative damage may be the reason behind age-dependent risk for developing primary open angle glaucoma. Chronic and acute effects of hydrogen peroxide (H2O2) on TM endothelial cells include changes in viability, protein synthesis, and cellular adhesion. However, little if anything is known about the immediate effect of H2O2 on the biochemistry of the TM cells and the initial response to oxidative stress. In this report, we have used two-photon excitation autofluorescence (2PAF) to monitor changes to TM cell nicotinamide adenine dinucleotide (NADPH). 2PAF allows non-destructive, real-time analysis of concentration of intracellular NADPH. Coupled to reduced glutathione, NADPH, is a major component in the anti-oxidant defense of TM cells. Cultured human TM cells were monitored for over 30 min in control and H2O2-containing solutions. Peroxide caused both a dose- and time-dependent decrease in NADPH signal. NADPH fluorescence in control and in 4 mM H2O2 solutions showed little attenuation of NADPH signal (4% and 9% respectively). TM cell NADPH fluorescence showed a linear decrease with exposure to 20 mM H2O2 (-29%) and 100 mM H2O2 (37%) after a 30 min exposure. Exposure of TM cells to 500 mM H2O2 caused an exponential decrease in NADPH fluorescence to a final attenuation of 46% of starting intensity. Analysis of individual TM cells indicates that cells with higher initial NADPH fluorescence are more refractive to the apparent loss of viability caused by H2O2 than weakly fluorescing TM cells. We conclude that 2PAF of intracellular NADPH is a valuable tool for studying TM cell metabolism in response to oxidative insult. (C) 2011 Elsevier Ltd. All rights reserved.
PD SEP
PY 2011
VL 93
IS 3
SI SI
BP 316
EP 320
DI 10.1016/j.exer.2011.02.012
ER
Author Response
1) the authors should explain the reason why, in the ONC model, a dose of NAD+ was given to the mice one day prior to the procedure. Wouldn’t the damage to retinal ganglion cells be greater if the animals had not received this?
Thank you for your comment; it led us to improve our explanation for experimental rationale (lines 24, 70, and 84, and mentioned in Discussion in lines 524-525 and 572). The rationale for giving a dose of NR prior to ONC was to ensure that effects on retinal NAD+ elevation existed prior to insult. This was done in order to mimic conditions that exist with ongoing and expected human use. NR has been successfully reviewed by the US FDA as a generally recognized as safe (GRAS) new dietary ingredient (NDI) and has similar regulatory classifications with the European Union and Australia. As a nutritional dietary supplement, humans take NR prophylactically, similar to how vitamin D, magnesium, or other supplements are taken. Thus, dosing prior to ONC is relevant for preclinical assessment.
We agree that it is likely that damage to retinal ganglion cells would be greater if the animals had not received an NR dose one day prior to ONC. Although we did not test this in the current ONC model, our group tested prophylactic dosing versus post-injury dosing alone in a light-induced retinal degeneration mouse model and found significantly more protection with prophylactic dosing. These data were published in Zhang et al., 2020, PMID:32852543.
2) figure 1 shows that after the same 5 days NAD+ pretreatment, the levels of NAD+ in retinal homogenates are considerably different for the two mouse strains, with increase of only ~17% for DBA2J and ~205% for C57BL/6J. Is there an explanation for that? The authors should at least mention this finding in the discussion section.
Not discussing this strain difference was an oversight on our part and we thank you catching this error. The animals were handled identically, so we currently have no explanation for the differences other than to suggest genetic background differences. We agree with your suggestion to discuss, though, and have accordingly added text on line 529 to 540.
Minor points:
-lines 302 and 305: correct typo…. Through.
Thank you for your comment. We replaced the word “trough” with “nadir” throughout the manuscript (lines 173, 174, 285, 318 and 321).
-line 369-370:….ILM…. OPL…correct the abbreviatures to match those shown in the figure
Thank you for catching those errors. We corrected them.
-line 404: please rephrase, as it is not clear what the authors meant to say
Thank you for the suggestion. We rephrased the sentence. Please see line 440-441 in the revised manuscript.
-lines 404: what does OHT stand for? ocular hypertension? it should be defined it at its first mention
Thank you for finding that omission. OHT is “ocular hypertension” and is now defined at line 443.
-412: correct typo …. versus
Thank you for catching the typo; it is corrected (now in line 449).
A list of relevant articles not cited in the manuscript follows:
Thank you for helping us properly cite the literature. The suggested references were added as citations 22, 24, and 25.

Reviewer 2 Report
Zhang et al have studied the neuroprotective effect of nicotinamide riboside (NR) in acute and chronic animal models of retinal ganlion cell damage. The introductory paragraph sets well the scenario under which the research work has been carried out and the methodology used to the scope are described in detail. The discussion seems to be, at least in part, supported by the results yielded. This referee has no major criticisms; however, there are some important aspects in the experimental design that hamper the impact of the data and should be taken into consideration by the Authors.
The number of animals used is quite important but the Authors do not provide justification for that nor they provide a sample power calculation to meet with the 3R requirements for animal studies. Also there is no blindness declared in the study. These two conditions make the preclinical data not rigorous enhough to meet with the quality requirement by instruments of current use for assessment of preclinical evidence.
The treatment schedule with NR comprehends a pretreatment phase for both experimental models that obviously limits the interpretation and translational value of the results. If the pretreatment phase is important for the whole set of neuroprotective results then the data are less important. In other words, the Authors should have demonstrated neuroptotection for NR tested in post-treatment. The functional, electrophysiological, data seem to support the neuroprotection afforded by NR. However, the data referred to the effects of NR on P1 and N2 do not seem to be reflected in the typical example reported in the Fig 1A and Fig 1B whilst the histograms report a more evident effect. This should be clarified because from the quality assessment of the electrophysiological responses reported in A and B the observer can only see a delayed recovery of the N2 wave but no dramatic chnges in the amplitude of P1 and less so for N2.
Whilst the method for measurement of RGC neuroprotection afforded by NR has been described accurately and, indeed, the neuroprotection afforded seems obvious this is at variance with the sampling for IOP assessment. It is not clear why the Author have measured the AUC for naive animals IOP but not the AUC of IOP of those animals treated with NR, regardless of the contraddictory result yieldes by NR and discussed by the Authors.
Author Response
- The number of animals used is quite important but the Authors do not provide justification for that nor they provide a sample power calculation to meet with the 3R requirements for animal studies. Also, there is no blindness declared in the study. These two conditions make the preclinical data not rigorous enough to meet with the quality requirement by instruments of current use for assessment of preclinical evidence.
Thank you for your observations and concerns. We apologize in advance if we misunderstand the first of the two comments, but we are not completely sure how to address a suggestion for a power analysis in what is now a post-hoc setting. As Chair of Emory University’s Institutional Animal Care and Use Committee (IACUC), I share your concerns about our responsibility to address “the 3Rs” rigorously. Our experiments were approved by Emory’s IACUC (PROTO201800248) prior to commencing, meeting the committee’s requirement for adequate power and the 3R requirement. To the second concern, the personnel conducting assessments in experiments that required judgement were masked to the specific treatment group from which sampling arose. This included semi-automated marking of pERG peaks and nadirs (Fig. 2), semi-automated counting of Brn3a-positive cells (Figs. 3 and 6), and counting GFAP-positive fibers (Fig. 4). Thank you for your kind reminder; we added text to explain these concepts in lines 100 to 101 and 282-286.
- The treatment schedule with NR comprehends a pretreatment phase for both experimental models that obviously limits the interpretation and translational value of the results. If the pretreatment phase is important for the whole set of neuroprotective results then the data are less important. In other words, the Authors should have demonstrated neuroprotection for NR tested in post-treatment.
Thank you for your comment; it led us to improve our explanation for experimental rationale (mentioned now in lines 24, 70, and 84, and mentioned in Discussion now in lines 524-525 and 570). The rationale for giving a dose of NR prior to ONC was to ensure that effects on retinal NAD+ elevation existed prior to insult. This was done in order to mimic conditions that exist with ongoing and expected human use and thus is translationally relevant. NR has been successfully reviewed by the US FDA as a generally recognized as safe (GRAS) new dietary ingredient (NDI) and has similar regulatory classifications with the European Union and Australia. As a nutritional dietary supplement, humans take NR prophylactically, similar to how vitamin D, magnesium, or other supplements are taken. Thus, dosing prior to ONC is relevant for preclinical assessment.
- The functional, electrophysiological, data seem to support the neuroprotection afforded by NR. However, the data referred to the effects of NR on P1 and N2 do not seem to be reflected in the typical example reported in the Fig 1A and Fig 1B whilst the histograms report a more evident effect. This should be clarified because from the quality assessment of the electrophysiological responses reported in A and B the observer can only see a delayed recovery of the N2 wave but no dramatic changes in the amplitude of P1 and less so for N2.
Thank you for your comment. We apologize that our explanation for marking P1 and N2 pattern ERG amplitudes was unclear. We added a new panel A in revised Figure 2 to clarify the measurement of P1 and N2 amplitudes (as shown below).
For pERG, P1 amplitude was measured from the N1 nadir to the peak of P1, N2 amplitude was measured from the preceding P1 peak to the nadir of N2. As shown in panels B and C below, both P1 and N2 amplitudes in PBS+ONC (red trace) declined compare with Naïve (black trace). NR treatment (green trace) significantly preserved the RGC function by increasing P1 and N2 amplitudes. The revised Fig. 2 has the added panel A that shows P1 and N2 measurements, but for simplicity and clarity, the detailed markings shown below for panels B and C are not presented.
- Whilst the method for measurement of RGC neuroprotection afforded by NR has been described accurately and, indeed, the neuroprotection afforded seems obvious this is at variance with the sampling for IOP assessment. It is not clear why the Author have measured the AUC for naive animals IOP but not the AUC of IOP of those animals treated with NR, regardless of the contradictory result yields by NR and discussed by the Authors.
Thank you for your comment. We apologize in advance if we misunderstand the point, but we did measure the AUC of IOP for animals treated with NR only. This is derived from the longitudinal data shown as the brown trace in Fig. 5A and its quantification is represented by the brown bar in Fig. 5C. Fig. 5B shows an example of how we calculated IOP AUC. Here we used the data from naïve group, again, just as an example to illustrate how the AUC of the IOP was derived.

Round 2
Reviewer 2 Report
The revised Ms is an improvement